# Accuracy of ZedView, the Software for Three-Dimensional Measurement and Preoperative Planning: A Basic Study

**DOI:** 10.3390/medicina59061030

**Published:** 2023-05-26

**Authors:** Asami Nozaki, Norio Imai, Kazuhisa Funayama, Yoji Horigome, Hayato Suzuki, Izumi Minato, Koichi Kobayashi, Hiroyuki Kawashima

**Affiliations:** 1Division of Orthopedic Surgery, Department of Regenerative and Transplant Medicine, Graduate School of Medical and Dental Sciences, Niigata University, Niigata 951-8510, Japan; 2Division of Comprehensive Musculoskeletal Medicine, Graduate School of Medical and Dental Sciences, Niigata University, Niigata 951-8510, Japan; 3Division of Legal Medicine, Department of Community Preventive Medicine, Graduate School of Medical and Dental Sciences, Niigata University, Niigata 951-8510, Japan; 4Department of Orthopedic Surgery, Niigata Rinko Hospital, Niigata 950-0051, Japan; 5School of Health Sciences, Faculty of Medicine, Graduate School of Medical and Dental Sciences, Niigata University, Niigata 951-8510, Japan

**Keywords:** cross-sectional anatomy, three-dimensional imaging, surgical navigation systems, CT-based planning, surgical-assistive software, presurgical planning

## Abstract

*Background and Objectives*: In the field of orthopedic surgery, novel techniques of three-dimensional shape modeling using two-dimensional tomographic images are used for bone-shape measurements, preoperative planning in joint-replacement surgery, and postoperative evaluation. ZedView^®^ (three-dimensional measurement instrument and preoperative-planning software) had previously been developed. Our group is also using ZedView^®^ for preoperative planning and postoperative evaluation for more accurate implant placement and osteotomy. This study aimed to evaluate the measurement error in this software in comparison to a three-dimensional measuring instrument (3DMI) using human bones. *Materials and Methods*: The study was conducted using three bones from cadavers: the pelvic bone, femur, and tibia. Three markers were attached to each bone. Study 1: The bones with markers were fixed on the 3DMI. For each bone, the coordinates of the center point of the markers were measured, and the distances and angles between these three points were calculated and defined as “true values.” Study 2: The posterior surface of the femur was placed face down on the 3DMI, and the distances from the table to the center of each marker were measured and defined as “true values.” In each study, the same bone was imaged using computed tomography, measured with this software, and the measurement error from the corresponding “true values” was calculated. *Results*: Study 1: The mean diameter of the same marker using the 3DMI was 23.951 ± 0.055 mm. Comparisons between measurements using the 3DMI and this software revealed that the mean error in length was <0.3 mm, and the error in angle was <0.25°. Study 2: In the bones adjusted to the retrocondylar plane with the 3DMI and this software, the average error in the distance from the planes to each marker was 0.43 (0.32–0.58) mm. *Conclusion*: This surgical planning software could measure the distance and angle between the centers of the markers with high accuracy; therefore, this is very useful for pre- and postoperative evaluation.

## 1. Introduction

The development of three-dimensional (3D) shape models from two-dimensional tomographic images became possible recently, and these models allow for the observation of physical internal structures of the body in three dimensions. In the field of orthopedic surgery, these techniques are used to form 3D models of bones and have been clinically applied to the measurement of bone shape, preoperative planning in joint-replacement surgery, and postoperative evaluation [1,2,3,4]. In addition to their use for diagnosis and treatment, these models are also being used for educational purposes [5,6].

Planning software is used to create 3D models for surgery, and 3D evaluations of bone morphology can allow more accurate implant positioning as well as postoperative verification [7,8]. Although X-ray image-based planning software programs are also available, they are not sufficiently accurate, with some reports suggesting that the measurements with these software programs had an average error of more than 7° [9]. Poor implant placement results in implant instability, wear [10], dislocation, and limited range of motion [11], which may affect treatment outcomes and patient satisfaction [12]. Thus, accurate implant placement is important, and 3D-computed tomography (CT) templates are considered to be effective for this purpose. Our group has been using ZedView^®^, a 3D measurement and preoperative planning software, for preoperative planning and postoperative evaluation for more accurate implant placement and osteotomy [13,14,15,16,17,18].

ZedView^®^ includes a 3D template system for implantation, such as for joint replacement (hip, knee, and shoulder joints) and posterior spinal fusion, as well as a 3D template system for osteotomy, fractures, and trauma. In addition, it also includes 3D construction and measurement software and 3D dynamic-analysis software. The software can perform various preoperative planning procedures, simulate postoperative bone movements during such planning, and perform three-dimensional measurements and evaluations of postoperative implant placement, angulation, and alignment, as well as rotation angles after osteotomy [13,19,20]. Although the measurement accuracy of this surgical-planning software has been evaluated using model bones [21], none of the previous studies evaluated its accuracy using human bones. Thus, this study aims to evaluate the measurement error in this surgical planning software, using a 3D measuring instrument (3DMI) on dry human bones as a reference, for the first time. 3DMI is a stationary measuring machine that applies a contactor to an object on a table and obtains three-dimensional coordinates from the object’s length, width, and height, enabling high-precision measurement of the object’s dimensions, positioning, contour shape, and geometric tolerances.

## 2. Materials and Methods

This study was conducted using three limbs: the pelvic bone, femur, and tibia of one side. The bones were obtained from skeletons (age and sex unknown) collected by the Department of Forensic Medicine, Niigata University between 1980 and 2020, which were in good condition and had all three bones. The three limbs were defined as Subjects 1, 2, and 3. As shown in Figure 1, three spherical markers with diameters of almost 24 mm, were attached to the pelvic bone, femur, and tibia. The femur and tibia are reproducibly stable when placed with the condyle down, but in the case of the unilateral ilium, the coordinates of the spherical marker from the platform changed depending on how the ilium was placed, so the pelvic bone was fixed to a wooden board with the quadrilateral face down and the spherical marker fixed to the outer surface to ensure that the coordinates did not change. The bones with these spherical markers attached were used to test accuracy by comparing measurement data from 3DMI with data reconstructed with ZedView from CT. The 3DMI was a UPMC550 (Carl Zeiss, Jena, Germany). The measurements using the 3DMI are shown in Figure 2. A multi-slice CT scanner with a 64-row detector (Aquilion 64TM, Toshiba Medical Systems, Otawara, Tochigi, Japan) was used to acquire approximately 600 slices (slice thickness, 1.25 mm) from each limb.

This investigation comprised the following two studies (study 1 and 2). In both studies 1 and 2, measurements were performed by three observers (observer A, B, and C). Observer A performed each measurement twice and assessed intra-observer error. Inter-observer error was also assessed using the results of the first measurement of observer A and the measurements of observers B and C.

### 2.1. Study 1

In this study, we used each of the bones of Subjects 1, 2, and 3. The dry bones with their attached spherical markers were positioned as follows: each femur and tibia was placed on the horizontal table of the 3DMI with their posterior surfaces facing downwards and then fixed directly to the table, while each pelvic bone was first fixed to a wooden board, which was then fixed directly to the 3DMI. For this study, the femur was set on the same plane as the retrocondylar plane, which was composed of the most posterior point of the medial and lateral posterior condyles and the most posterior point of the greater trochanter [22]. For each bone fixed on the 3DMI platform, the coordinates of the center points of the three spherical markers were measured using the 3DMI. The distance and angle between the three points were then calculated from the coordinates of the three points (Figure 3). The values calculated from these 3DMI-measured coordinates are the “true values”. The same dry bone was imaged using CT; the distance and angle between the centers of the sphere markers were measured using this surgical-planning software, and the measurement error from the corresponding “true value” was calculated.

### 2.2. Study 2

In this study, we used only the femurs of Subjects 1, 2, and 3. The posterior surface of each femur was placed face down on the horizontal surface of the table of the 3DMI and fixed. The distance from the horizontal surface of the table to the center of each sphere marker was measured (Figure 4). At this time, the femur was placed on the same plane as the retrocondylar plane. The same dry bones were imaged using CT; the femurs were adjusted to the retrocondylar plane in this surgical-planning software. The retrocondylar plane is the plane consisting of the posterior margin of the greater trochanter and the posterior condyles of both femurs, which allows the 3DMI platform with the femur fixed to be reproduced when measuring with the 3DMI. The distances between the retrocondylar plane and the centers of the sphere markers were measured in the same manner, and the measurement error from the “true values” was calculated.

### 2.3. Statistical Analysis

We used IBM SPSS Statistics for Windows (version 24.0; IBM Corp., Armonk, NY, USA) to analyze the data. To assess the error from the true values as well as the intra- and interobserver error, we evaluated the mean absolute difference (MAD) and variability by determining the standard deviation. In addition, the reliability of the values measured using this surgical planning software was evaluated with interclass correlation coefficient (ICC) values to verify intra- and interobserver reliability. ICCs greater than 0.8 were considered reliable.

This study was designed in accordance with the principles of the Declaration of Helsinki and approved by the Niigata University Institutional Review Board (2020-3048). The need to obtain informed consent was waived because this was experimental research intended to evaluate surgical assistive software, so no interventions were performed, and the participants could not be identified.

## 3. Results

### 3.1. Study 1

The mean diameter of the spherical marker after nine repeated measurements with the 3DMI was 23.95 ± 0.06 mm (range: 23.87–24.04 mm), and the maximum measurement error was 0.17 mm. The measurement error between the distance and angle measurements between the pelvic bone, femur, and tibia markers by each observer and the 3DMI measurements is shown in Table 1. The mean error between 3DMI and observer A (1st) was 0.17 ± 0.15 mm and 0.12 ± 0.07° for the pelvis bone, 0.21 ± 0.09 mm and 0.04 ± 0.04° for the femur and 0.21 ± 0.17 mm and 0.07 ± 0.04° for the tibia. Similarly, the mean error between 3DMI and observer A (2nd) was 0.14 ± 0.12 mm and 0.08 ± 0.07° for the pelvis bone, 0.20 ± 0.10 mm and 0.09 ± 0.07° for the femur and 0.26 ± 0.21 mm and 0.25 ± 0.17° for the tibia. The mean error between 3DMI and observer B was 0.23 ± 0.17 mm and 0.20 ± 0.13° for the pelvis bone, 0.18 ± 0.14 mm and 0.16 ± 0.08° for the femur, and 0.14 ± 0.09 mm and 0.12 ± 0.09° for the tibia. The mean error between 3DMI and observer C was 0.15 ± 0.08 mm and 0.20 ± 0.14° for the pelvis bone, 0.21 ± 0.15 mm and 0.11 ± 0.04° for the femur and 0.10 ± 0.08 mm and 0.06 ± 0.04° for the tibia. The average errors in length were less than 0.3 mm, and the maximum errors were 0.66 mm, 0.54 mm, and 0.54 mm for the pelvis, femur, and tibia, respectively. The average errors in angle were less than 0.25°, and the maximum errors for the pelvis, femur, and tibia were 0.40°, 0.31°, and 0.49°, respectively. For all pelvic bone, femur, and tibia, the interclass correlation coefficient (ICC) for length was 1.00, both within and between examiners; the ICC for angle was also 1.00. The *p*-values were also <0.001 for all of the above. This indicates that the reliability of the test values was high in all the assessments.

### 3.2. Study 2

In the measurement of the distance from the posterior condylar plane to each spherical marker, the measurement error between the measurements in the 3DMI and the measurements in this surgical-planning software by each observer is shown in Table 2. The mean error between 3DMI and observer A (1st) was 0.32 ± 0.25 mm. Similarly, the mean error between 3DMI and observer A (2nd) was 0.58 ± 0.37 mm. The mean error between 3DMI and observer B was 0.34 ± 0.21 mm. The mean error between 3DMI and observer C was 0.46 ± 0.34 mm. The average error in the distance from the examination table and the retrocondylar plane to each sphere marker was 0.21 to 0.75 mm, and the maximum error was 1.10 mm (Table 2). The reliabilities of the test values were high, with a mean intraobserver ICC value of 0.999 (0.997–1.000, *p* < 0.001) and an interobserver ICC value of 0.999 (0.998–1.000, *p* < 0.001).

## 4. Discussion

ZedView^®^, surgical-planning software developed by Lexis, Inc. to create, display, and measure 3D models from 2D continuous slice images, is currently widely used. In addition to measuring the distance and angle between two points [18,19], it can also measure areas and volumes [23,24]. Depending on their application, they are divided into software such as ZedHip^®^, ZedKnee^®^, ZedShoulder^®^, ZedSpine^®^, JIGEN^®^, ZedOsteotomy^®^, and ZedTrauma^®^. There is also ZedEdit^®^, which allows 3D images to be adjusted in any plane and 2D images to be created from 3D images [16]. In recent years, AI-assisted functions for the automatic detection of anatomical reference points and construction of 3D models with automatic segmentation and 3D-3D matching have also been included. In addition, the created 3D model in ZedMotion^®^ can be applied to various fields such as joint range of motion simulation [25]. Although accuracy tests have been reported on this software using model bones and animal bones, this is the first study using human bones [9,26].

In study 1, the mean error of the measurements of the distance between the sphere marker centers using this software was less than 0.3 mm and the maximum error was 0.66 mm with reference to the 3DMI measurements. The mean error of the measurements of the spherical marker centers’ angle using this software was less than 0.3° and the maximum error was 0.49°. Since both were less than 1 mm and 1°, respectively, this software can be considered to be as accurate as a “line gauge.” There have been previous reports testing the accuracy of implant-placement angles [27] and postoperative leg-length differences [28], but these have mainly been reported in 1.0° and 1.0 mm increments. Therefore, the mean error in this study was 0.3° and 0.3 mm, and such an error of less than 1° and 1 mm is sufficiently accurate. This result was similar to that obtained in a similar survey conducted on the femur of a pig cadaver [9]. Sasagawa verified the accuracy of 3D bone-surface models by performing MRI and CT evaluations of the femur of pig cadavers, and the average error in CT-based measurements was approximately 0.4 mm [26]. In addition, Imai et al. reported that the intra-examiner error in the measurement of the anteversion angle and the inclination angle of the acetabular component after THA in the pelvis adjusted to the anterior pelvic plane was 0.56° ± 0.70° and 0.65° ± 0.67°, respectively, and the inter-observer error was 0.62° ± 0.67° and 1.10° ± 1.15°, respectively [17]. Similar results were obtained in the present study.

In study 2, the outlines of the bones were used, adjusted to the retrocondylar plane in this software, and compared with 3DMI measurements. The error was 0.21–0.58 mm, and the maximum error was 1.10 mm. In a previous study using this software, the femur was adjusted to the posterior condylar plane, and the intraobserver error for the measurement of the distance between the trans-epicondylar axis and the most distal point of the medial posterior condyle was 0.73 ± 0.52 mm, while the interobserver error was 0.86 ± 0.77 mm [26]; this was similar to the results of this study. These results suggested that when bone outlines were used and the plane was adjusted, the lack of clarity of the bone contour affected the adjustment of the plane and the determination of the reference point, resulting in a measurement error of approximately 1 mm.

The major limitation of this study was the small number of subjects. The collection of measurable dry bones that retain the shape of the entire bone is challenging; therefore, this may be an unavoidable limitation. In addition, different CT images show different detection accuracies [29]. Therefore, the results of this study can only be applied to the same CT setup. However, this study was not conducted to compare this surgical-planning software with other computer programs, so if there is no significant difference in the quality of the CT images, similar results can be obtained. Ideally, measurements should be taken on the actual patient during the operation to verify the accuracy. However, it was not possible to measure the actual patient on the 3D measuring device intraoperatively, as was carried out in the measurements performed in this study. Therefore, it was difficult to accurately measure in vivo intraoperatively and verify the accuracy, which is considered to be one of the limitations.

## 5. Conclusions

This study was the first to test the accuracy of this surgical-planning software using human bones as far as we are aware, although there have been some investigations due to bone models and other factors. This software was able to measure the distance and angle between the centers of the spherical markers with high accuracy. When the outlines of bones were used to adjust the plane, the error was expected to be approximately 1 mm; hence, errors within 1 mm were considered unremarkable. Therefore, we conclude that this surgical-planning software is very useful for preoperative planning, postoperative evaluation, and 3D alignment evaluation with sufficiently high accuracy.

## Figures and Tables

**Figure 1 medicina-59-01030-f001:**
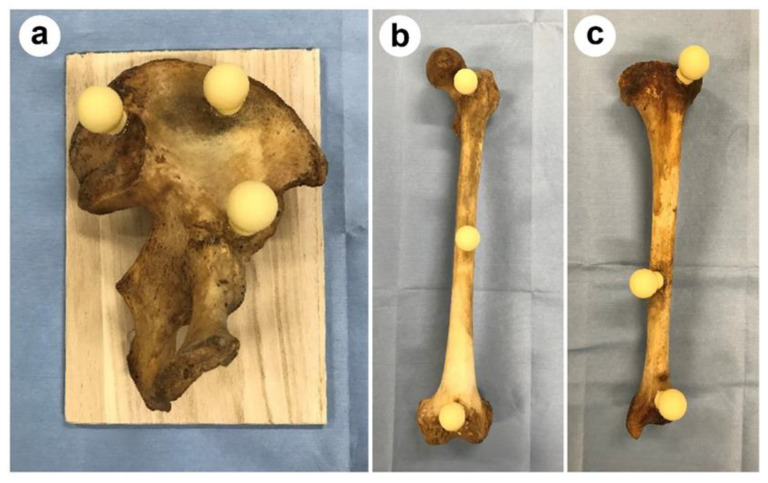
Three spherical markers were attached as shown here to each bone. (**a**) the pelvic bone, (**b**) femur, and (**c**) tibia.

**Figure 2 medicina-59-01030-f002:**
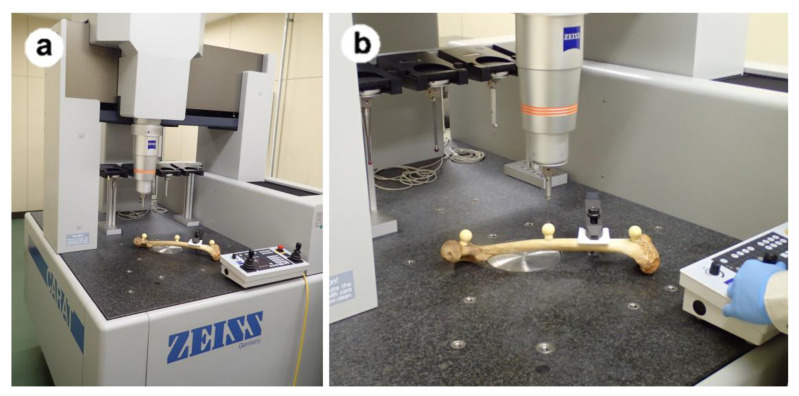
Each bone was fixed and measured by 3DMI. (**a**) Overall, (**b**) During measurement.

**Figure 3 medicina-59-01030-f003:**
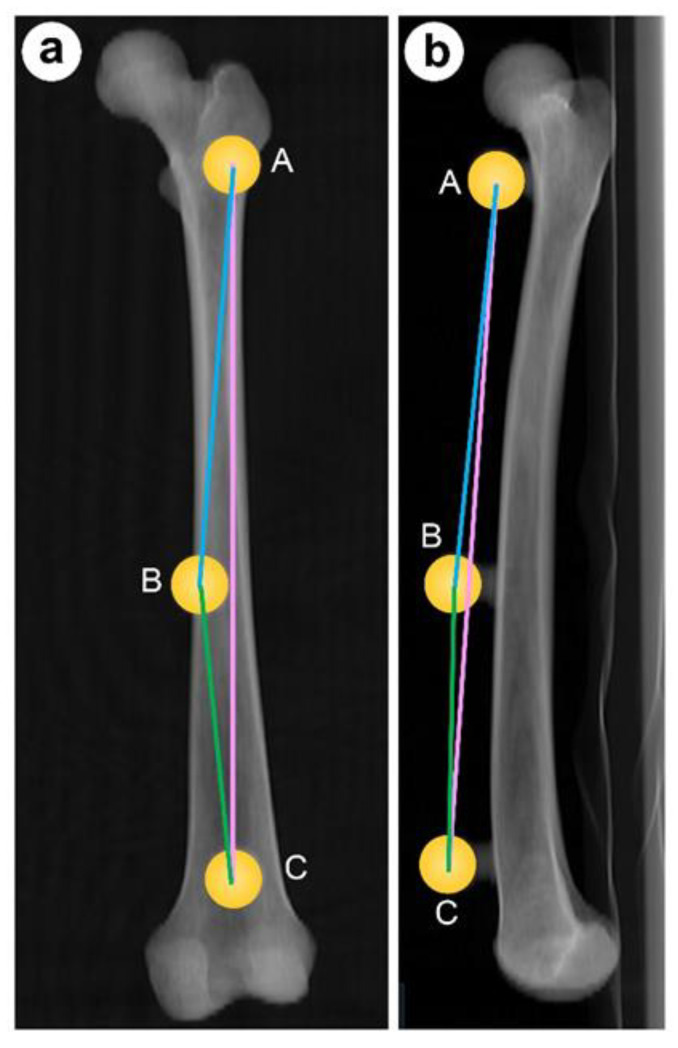
In study 1, the coordinates of the center points of the spherical markers were determined, and the distances and angles between these three points were calculated. As an example, the femur is shown here in (**a**) frontal and (**b**) lateral view.

**Figure 4 medicina-59-01030-f004:**
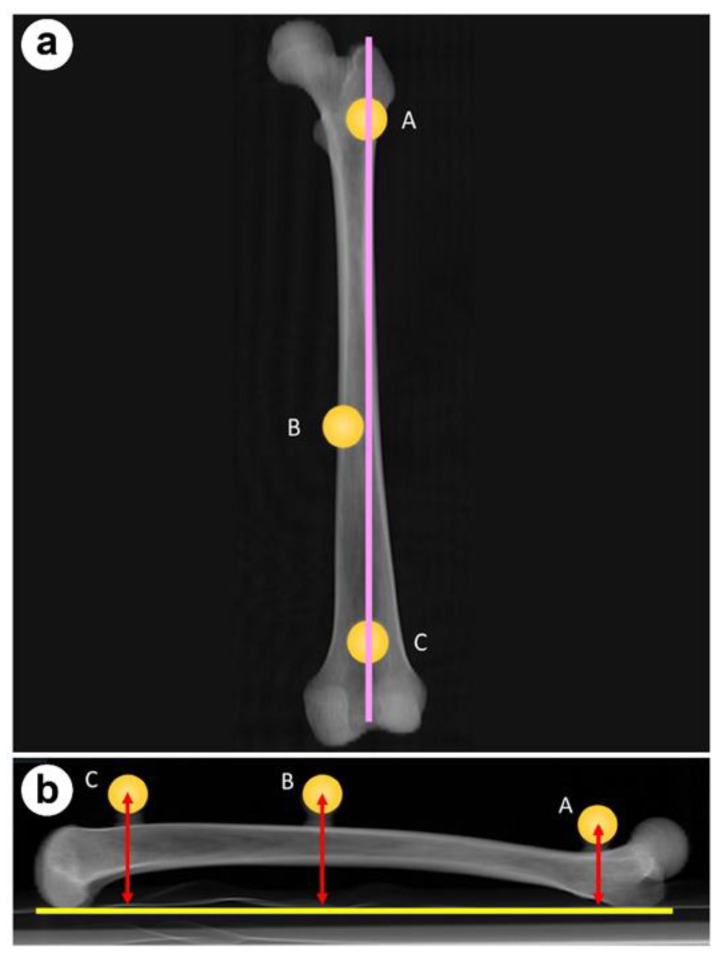
In study 2, the distance between the center of each spherical marker on the femur and the horizontal surface of the table, represented by the yellow line shown in (**b**), was measured. This image shows the (**a**) frontal and (**b**) lateral views.

**Table 1 medicina-59-01030-t001:** Comparisons of the measurement errors among the three-dimensional measuring instrument (3DMI) and each observer.

		3DMI— Observer A 1st	3DMI— Observer A 2nd	3DMI— Observer B	3DMI— Observer C
Length	Pelvis bone (mm)	0.17 ± 0.15 (0.003~0.36)	0.14 ± 0.12 (0.004~0.38)	0.23 ± 0.17 (0.05~0.66)	0.15 ± 0.08 (0.02~0.28)
	Femur (mm)	0.21 ± 0.09 (0.12~0.38)	0.20 ± 0.10 (0.04~0.32)	0.18 ± 0.14 (0.004~0.45)	0.21 ± 0.15 (0.05~0.54)
	Tibia (mm)	0.21 ± 0.17 (0.02~0.49)	0.26 ± 0.21 (0.01~0.54)	0.14 ± 0.09 (0.02~0.32)	0.10 ± 0.08 (0.03~0.25)
Angle	Pelvis bone (°)	0.12 ± 0.07 (0.04~0.25)	0.08 ± 0.07 (0.004~0.20)	0.20 ± 0.13 (0.04~0.40)	0.20 ± 0.14 (0.008~0.40)
	Femur (°)	0.04 ± 0.04 (0.008~0.11)	0.09 ± 0.07 (0.008~0.23)	0.16 ± 0.08 (0.08~0.31)	0.11 ± 0.04 (0.07~0.21)
	Tibia (°)	0.07 ± 0.04 (0.01~0.15)	0.25 ± 0.17 (0.04~0.49)	0.12 ± 0.09 (0.004~0.29)	0.06 ± 0.04 (0.005~0.11)

Upper row: mean absolute difference ± standard deviation; lower row: range.

**Table 2 medicina-59-01030-t002:** Measurement errors between coordinate measurements were obtained using the machine and by each observer.

	3DMI— Observer A 1st	3DMI— Observer A 2nd	3DMI— Observer B	3DMI— Observer C
Femur of Subject 1 (mm)	0.21 ± 0.28 (0.009–0.53)	0.60 ± 0.40 (0.27–1.04)	0.30 ± 0.07 (0.24–0.37)	0.75 ± 0.49 (0.36–1.31)
Femur of Subject 2 (mm)	0.52 ± 0.28 (0.25–0.81)	0.61 ± 0.53 (0.05–1.10)	0.35 ± 0.29 (0.08–0.66)	0.37 ± 0.14 (0.26–0.53)
Femur of Subject 3 (mm)	0.24 ± 0.07 (0.17–0.31)	0.51 ± 0.30 (0.22–0.82)	0.38 ± 0.28 (0.05–0.58)	0.25 ± 0.05 (0.21–0.30)
Total (mm)	0.32 ± 0.25 (0.009–0.81)	0.58 ± 0.37 (0.05–1.10)	0.34 ± 0.21 (0.05–0.66)	0.46 ± 0.34 (0.21–1.31)

Upper row: mean absolute difference ± standard deviation; lower row: range.

## Data Availability

All data generated or analyzed during this study are included in this published article.

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
