# Peer review of "Accuracy of ZedView, the Software for Three-Dimensional Measurement and Preoperative Planning: A Basic Study"

_medicina, 2023, doi:10.3390/medicina59061030_

Round 1
Reviewer 1 Report
1. Lines 31: Please use two decimal places to express numbers consistently throughout the article.
2. Please cite the reference based on the journal's style.
3. Please make a short paragraph describing this device and try to focus on what kinds of orthopedic surgery. Joint replacement or corrective osteotomy of bone deformity?
4. Line 71: Please mention the diameter of the spherical markers.
5. Line 91-94: Since the spherical markersprettyquite big, please describe how to measure the "true value" at CT. Do you repeat it three times and calculate the mean?
6. Line 141: Since the ICC is all 1, it is unnecessary to present the data in a table.
7. Line 193: It is improper to state that it is the "first."
8. Lines 165-166 The accuracy is based on what scale of the object you are measuring. Please cite the appropriate reference to make the statement < 1 mm and < 1° are considered to be accurate.
9. Lines 194-195: what is high accuracy based on?
Author Response
Reviewer #1.
Thank you for your invaluable feedback. We have now revised our paper based on your comments, which we believe has significantly improved the quality of our paper. Our responses to your comments are provided below.
Comments and Suggestions for Authors
- Lines 31: Please use two decimal places to express numbers consistently throughout the article.
Answer: Thank you for pointing this out. The text has been corrected.
- Please cite the reference based on the journal's style.
Answer: Thank you for pointing this out. The references have been corrected.
- Please make a short paragraph describing this device and try to focus on what kinds of orthopedic surgery. Joint replacement or corrective osteotomy of bone deformity?
Answer: Thank you for pointing this out. We have added it to the text.
- Line 71: Please mention the diameter of the spherical markers.
Answer: The diameter of the spherical markers was 24 mm. I have added to the text.
- Line 91-94: Since the spherical markersprettyquite big, please describe how to measure the "true value" at CT. Do you repeat it three times and calculate the mean?
Answer: We defined "true values" as the values measured by 3DMI. The true values are compared with the angles and distances measured from the CT images using Zedview. The description was not clear and has been corrected.
- Line 141: Since the ICC is all 1, it is unnecessary to present the data in a table.
Answer: Thank you for pointing this out. I have corrected.
- Line 193: It is improper to state that it is the "first."
Answer: Although there have been reports on bone models and animal bones, none have been investigated on human bones, which is why we described this as the first report of an investigation on human bones. A few expressions were added.
- Lines 165-166 The accuracy is based on what scale of the object you are measuring. Please cite the appropriate reference to make the statement < 1 mm and < 1° are considered to be accurate.
Answer: Thank you for pointing this out. There have been reports verifying the accuracy of implant placement angles and post-operative leg length differences, but mainly in units of 1.0° and 1.0 mm. Therefore, we consider that an error of 1° or 1 mm or less is sufficient accuracy. The above has been added to the text using citations.
- Lines 194-195: what is high accuracy based on?
Answer: Thank you for pointing this out. We considered it as a response to your point #8 and changed it to high accuracy, but we have changed it to sufficiently high accuracy.
Reviewer 2 Report
Congratulations on such an innovative paper regarding a software tool and method that, despite is huge value and advantage offered in real clinical setup, it is not yet widely adopted by orthopedic surgeons worldwide.
Title: I would suggest removing “the” from the title and just use “a” to reference the software in question, since this is not the only tool available (see comments below, for lines 60-66).
Lines 60-66: Please specify the reason for selecting ZedView software suite, in contrast with other platforms (such as MIMICS, for example). Does it provide better overall results? Ease of use? Additional features?
Lines 185-191: Please also add to the limitations that your study assess just the accuracy of a software tool in an in vitro situation and not with live specimens in an in-vivo setup, such as measuring preoperatively the patient with the provided tool and then having the same measurements during the surgical procedure, for comparison.
Author Response
Reviewer #2.
Thank you for your invaluable feedback. We have now revised our paper based on your comments, which we believe has significantly improved the quality of our paper. Our responses to your comments are provided below.
Congratulations on such an innovative paper regarding a software tool and method that, despite is huge value and advantage offered in real clinical setup, it is not yet widely adopted by orthopedic surgeons worldwide.
Title: I would suggest removing “the” from the title and just use “a” to reference the software in question, since this is not the only tool available (see comments below, for lines 60-66).
Answer: Thank you for your feedback. We have only verified the ZedView in this study, so we have assumed it to be "the".
Lines 60-66: Please specify the reason for selecting ZedView software suite, in contrast with other platforms (such as MIMICS, for example). Does it provide better overall results? Ease of use? Additional features?
Answer: The main reason is that ZedView is the software we normally use. We have used ZedView for a variety of measurements in the past, but we were sometimes asked about the accuracy of ZedView itself, so we conducted a survey on ZedView this time.
Lines 185-191: Please also add to the limitations that your study assess just the accuracy of a software tool in an in vitro situation and not with live specimens in an in-vivo setup, such as measuring preoperatively the patient with the provided tool and then having the same measurements during the surgical procedure, for comparison.
Answer: Thank you for pointing this out. It is true that the point you pointed out is Limitation. However, in this study, we examed the accuracy of the ZedView by comparing it with a 3D measuring device, and it is not possible to actually measure a patient with a 3D measuring device intraoperatively. Therefore, we consider it difficult to verify the accuracy by measuring accurately intraoperatively. We describe this as a Limitation.